# Stress Induces Trait Variability across Multiple Spatial Scales in the Arid Annual Plant *Anastatica hierochuntica*

**DOI:** 10.3390/plants13020256

**Published:** 2024-01-16

**Authors:** Nir Krintza, Efrat Dener, Merav Seifan

**Affiliations:** 1Albert Katz International School for Desert Studies, Jacob Blaustein Institutes for Desert Research, Ben-Gurion University of the Negev, Midreshet Ben-Gurion 8499000, Israel; nir.krintza@gmail.com; 2Mitrani Department of Desert Ecology, Swiss Institute for Dryland Environment and Energy Research, Jacob Blaustein Institutes for Desert Research, Ben-Gurion University of the Negev, Midreshet Ben-Gurion 8499000, Israel; efratde@gmail.com

**Keywords:** deserts, phenotypic plasticity, functional traits, evolutionary ecology, plant–climate interactions

## Abstract

Variations in plant characteristics in response to habitat heterogeneity can provide valuable insights into the mechanisms governing plant responses to environmental conditions. In this study, we investigated the role of environmental factors associated with arid conditions in shaping the phenotypic responses of an arid annual plant, *Anastatica hierochuntica*, across several populations found along an aridity gradient and across multiple spatial scales. Utilizing both field surveys and a net house experiment, we assessed the effects of environmental factors on trait variability within and between populations. The results indicated a significant convergence in plant height due to site aridity, reflecting growth potential based on abiotic resources. Convergence was also observed in the plant’s electrolyte leakage with aridity and in plant height concerning soil salinity at specific sites. Phenotypic plasticity was pivotal in maintaining trait variability, with plant height plasticity increasing with soil salinity, SLA plasticity decreasing with aridity, and leaf number plasticity rising with aridity. In conclusion, our findings underscore the adaptive significance of phenotypic variability, especially plasticity, in arid conditions. Notably, trait variability and plasticity did not consistently diminish in stressful settings, emphasizing the adaptive value of flexible responses in such environments.

## 1. Introduction

Understanding how plants respond to environmental conditions is a fundamental goal of ecological studies because the responses impact plant dynamics at all scales [1,2]. Variations in individual responses to heterogeneity in habitat conditions in space and time are often reflected in plant characteristics. If the affected characteristics can be considered as functional, differences in variation observed between individuals within and between sites can provide valuable insights into the governing mechanisms [2,3,4]. For example, an increase in mean environmental stress typically corresponds to a change in the mean values of relevant traits [5,6]. Since environmental conditions are constantly changing in space and time, plants need to cope with several environmental factors simultaneously. At the individual level, particularly when the source of heterogeneity in environmental conditions is temporal and not only spatial, it is advantageous to maintain a flexible strategy that enables each individual to respond efficiently to varying conditions [7]. At the population level, the individual’s ability to respond to the sum of environmental conditions is reflected in the level of variability in trait values around the mean [4,8].

Phenotypic plasticity is a common individual response to environmental heterogeneity. When adaptive, it represents an ideal response to changes as it enables individuals to rapidly produce an appropriate phenotype for each environmental condition [9,10]. The degree of plasticity may change across a species’ distribution range and environmental gradients due to exposure to different environmental pressures [7,11]. In particular, high environmental heterogeneity is expected to enhance plasticity, e.g., Refs. [8,12]. This is because maintaining flexible responses through variability in physiological and morphological traits related to growth and reproduction promotes long-term fitness stability for plant populations [6,13]. However, in cases where harsh environmental conditions lead to an increase in the metabolic and physiological costs associated with maintaining plasticity, the magnitude of plasticity may become restricted, e.g., Refs. [14,15,16]. This limitation is often accompanied by intensified selection pressure due to harsh conditions, and consequently to local adaptation, which further constrains trait variability [6,7]. Moreover, the accuracy of the individual response to changing conditions can also constrain phenotypic plasticity. In highly unpredictable habitats, relying on current environmental cues may not provide relevant information for future conditions, and responses may become nonadaptive [9]. In harsh habitats, these processes may be further intensified due to population isolation and limited population size [5,11]. Recent studies have emphasized that although contradicting in their outcome, mechanisms that reduce variability and those that maintain it often interact as part of the plant’s coping strategies, e.g., Refs. [11,17].

Extreme arid habitats provide a convenient setting for studying plant responses to changing environmental conditions at multiple spatial scales due to the combination of intense stress and high environmental heterogeneity they offer [18,19]. Plants inhabiting these habitats are required to invest in specialized structures and mechanisms that enable them to cope with periods of extreme resource limitations and high stress while maintaining their ability to rapidly utilize available resources when conditions become favorable, e.g., Refs. [19,20,21]. In this study, we utilize the conditions in the Negev Desert, which are characterized by short and unpredictable rainfall seasons and dry, hot summers. We focused on *Anastatica hierochuntica* L. (Brassicaceae), a winter annual with a relatively wide distribution range, that colonizes mainly Wadies and runnels throughout the old-world desert region [22,23]. The species was already studied as a model for adaptation to arid conditions, including its dormancy and dispersal mechanisms [23,24], tolerance to heat and salinity [25,26], and germination strategies, e.g., Ref. [27]. 

We investigated the role of environmental factors associated with arid conditions in differentiating populations of *A. hierochuntica* growing in arid environments. Through field observations and a net house experiment, we sought to determine (i) whether we could identify specific environmental factors that are strongly associated with differences in *A. hierochuntica* traits within and between populations. We predicted that if habitat conditions imposed strong developmental and physiological constraints on the plants, we would detect significant effects of these conditions on *A. hierochuntica* trait averages, particularly between populations. We further asked (ii) whether trait variability and flexible responses are dominant mechanisms to cope with harsh conditions. We predicted that individuals of *A. hierochuntica* would exhibit trait convergence in response to their local site conditions. Under this scenario, trait similarity between individuals growing in similar local conditions would be stronger than trait similarity within a population, regardless of their population of origin.

We further asked (iii) whether phenotypic plasticity plays a significant role in maintaining the observed pattern of traits. We predict that, in accordance with the theoretical arguments concerning trait variability patterns, we would observe a divergent pattern in the level of phenotypic plasticity across *A. hierochuntica* traits that are attributed to the selection pressure imposed by harsh environmental conditions and the heterogeneity in local conditions.

## 2. Results

### 2.1. Population Survey in the Field

The sampled individuals of all populations were similar in their range of traits, showing no significant effect of any of the habitat conditions measured on any of the chosen traits (Table 1, Appendix A). The only significant effect was the expected positive effect of plant height on the number of leaves (Figure 1). 

When analyzing the relationship between trait similarity and similarity in habitat conditions, more information was revealed (Table 2 and Figure 2). Plant height was significantly different between populations. Specifically, the values of plant height diverged at the site level so that the larger the differences in site aridity between populations, the larger the differences in plant height of the individuals growing in them (Figure 3). In addition, we detected a significant interaction term between similarity in aridity and similarity in salinity on plant height (Table 2). This interaction means that the height of plants growing in similar salt conditions within site, where aridity was the most similar, converged. When similarity in plant height was compared between sites, the more dissimilar the aridity values, the more dissimilar the plants’ height (Figure 3).

As in the case of plant height, electrolyte leakage values also diverge between sites, showing larger differences in leakage values between individuals growing in different sites relative to within sites (Figure 3).

### 2.2. Net House Experiment

As expected, the results revealed a significant effect of irrigation on plant height and plant biomass (Appendix A). The SLA also responded directly to the salinity in the soil of the mother plant, regardless of the irrigation treatment. In addition, we detected the following interaction terms between irrigation treatment and habitat conditions: plant height, leaf number, and biomass responded to an interaction between irrigation and salinity, and leaf number and the SLA responded also to an interaction between irrigation and aridity. The full information, tables, and figures can be found in the Appendix A. 

The effect of the habitat conditions of the mother plants at different scales on their offspring plasticity is easier to detect when analyzing the reaction norms directly in relation to habitat conditions (Table 3, Figure 4, and Appendix A). Using this analysis, we detected that plasticity in plant height responded to within-site soil salinity, with plants showing higher plasticity the more saline the local conditions in which their mother grew (Figure 5).

Plasticity in the number of leaves and in the SLA showed significant between-population effects, with plants showing a higher plasticity in leaf number and lower plasticity in the SLA the more arid the site (Figure 5).

Finally, the reaction norms of both plant biomass and fruit numbers did not show significant responses to any of the site conditions tested (Figure 5).

## 3. Discussion

Phenotypic changes provide a prevalent and effective mechanism for maximizing plant performance in the face of changing conditions [9,12,17]. This becomes especially crucial in arid regions, where conditions predominantly pose strong limitations, with occasional occurrences of favorable conditions [28]. In this study, we demonstrated that the outcome of such spatiotemporal variability is manifested as phenotypic variability both between and within populations. Although the general pattern of trait variation observed in the field was not easily decipherable, our analyses revealed distinct trait responses, particularly in plant height and electrolyte leakage, in relation to various stressors. Notably, previous studies have often associated the costs of stressful conditions with stronger constraints on phenotypic plasticity, particularly in morphological traits as measured here [7,29,30]. However, our findings indicate that the overall observed phenotypic variability, and phenotypic plasticity specifically, increased in response to stress factors across several scales. 

In the field, the observed phenotypic variation displayed distinct patterns of convergence in relation to habitat conditions, thus reflecting the capacity of local populations to withstand local environmental variability [18,30]. The most pronounced effect was observed in relation to plant height, which exhibited a significant response to aridity between sites. While investment in height is often regarded as an indicator of competitive ability, particularly in relation to light availability, e.g., Refs. [31,32], this interpretation is irrelevant in arid regions due to the relatively low density of most populations and the abundance of light and high radiation. Instead, in such arid regions, we propose that height represents the plant’s growth capacity in general, see also [12], and should be interpreted in relation to the effect of abiotic resources. This interpretation is supported by the finding that within site, plants that grew in similar saline conditions exhibited similar heights, suggesting a within-site convergent response to the level of salinity in the soil. A similar plant response has been documented also in other habitats, including salt marshes [33] and grasslands [34]. Furthermore, the plants’ sensitivity to abiotic conditions is supported by the similarity in electrolyte leakage within site. This indicates a convergence in the ability of the plants to maintain membrane integrity under harsh conditions, see also [35].

The observed phenotypic variation, and phenotypic plasticity in particular, have a well-established role in shaping population success. Nevertheless, there is still a lack of clear understanding regarding the causes of differences in plasticity between populations [29]. Moreover, only limited work has been conducted on the subject in arid regions, e.g., Refs. [12,36], even though arid regions are characterized by strong stress gradients and high spatiotemporal variation in conditions and offer convenient case studies [28]. Recently, Martinez-Vilalta et al. [37] highlighted that the challenges arise from two primary sources. First, the multifactor nature of the forces that affect population responses. Second, interactions between trait functions and between traits and environmental factors that cause the responses to be scale- and context-dependent [37].

In this study, we helped fill this gap by providing evidence for the contribution of phenotypic plasticity to maintaining trait variability in three of the studied traits.

We observed that plants growing in higher soil salinity, regardless of their origin, exhibited higher plasticity in plant height. These findings may be linked to enhanced and flexible responses under environmental stress, contributing to the maintenance of plant vigor, e.g., Ref. [33]. In contrast, plasticity in the SLA decreased with aridity between the chosen sites. This reduction in plasticity indicates the association between the SLA and resource conservation and stress tolerance, e.g., Refs. [32,38], which become increasingly crucial with site aridity. Unfortunately, we could not support this conclusion by directly testing electrolyte leakage, as we did in the field survey, due to pandemic restrictions. However, this conclusion is supported by other regional-scale studies that found that the costs of plasticity may be higher in unfavorable habitats, particularly for traits such as the SLA that directly impact the plant’s ability to respond to stress factors [39,40,41]. 

Interestingly, it is important to note that while plasticity in the SLA decreased, plants exhibited increased plasticity in the number of leaves they produced. This flexibility in leaf number can be interpreted as a compensatory mechanism to offset the limited adjustability of within-leaf mechanisms reflected in the lower SLA plasticity. This interpretation is supported by experimental findings based on Arabidopsis thaliana, where reproductive success was associated with either large SLA values and a small number of leaves or with a large number of leaves and small SLA values [42]. We additionally found a plastic response in leaf number in relation to soil salinity, with higher plasticity in plants that originated in more saline conditions. This fits previous findings that phenotypic variation in general, and variation in the number of leaves in particular, are strongly affected by habitat salinity, e.g., Ref. [43]. Plastic response to salinity was found to affect leaves by increasing respiration and reducing leaf area and limiting cell division, which generally limit growth, e.g., Refs. [44,45], but may also result in more leaves [46]. Together, the results further support the conclusion that *A. hierochuntica* relies on plastic responses in relation to harsh conditions. This conclusion aligns with a previous study conducted in an arid habitat, albeit different, linking variability in functional traits to harsher conditions and especially to drought [12].

Finally, it is interesting to note that we did not observe clear trends in the reaction norms produced by either the biomass or fruit set. The absence of significant shifts may suggest that although functional traits associated with growth exhibited phenotypic variation within and between populations, this variation did not provide adaptive advantages as it was not reflected in direct measures of fitness, such as the fruit set [10,11]. However, the lower phenotypic variation observed in fitness-related traits could also be interpreted as adaptive, because it may indicate that the plants possess the ability to compensate for harsh conditions and maintain a similar performance across different environmental conditions [29,47]. In line with this interpretation, we propose that the phenotypic variation in morphological traits related to plant growth exhibited significant, albeit occasionally challenging to decipher, sensitivity to diverse stress factors, thus serving as a supportive function toward the ultimate goal of plasticity—reducing fitness disparities in changing conditions [17]. 

Our results showed a remarkable consistency despite the variable conditions in which we conducted our study. Nevertheless, it is important to acknowledge certain technical issues that might have affected the robustness of the findings. Firstly, the sample size utilized in this study was relatively small. This limitation is common when working with wild flora in remote locations characterized by highly unpredictable conditions, and it may have affected the power of our statistical models. For example, initially, our data set was three times larger, comprising 45–50 plants per four field sites. Unfortunately, an unexpected heat wave resulted in the loss of our original populations before we could complete the sampling. In addition, the germination rate in the net house was 30–40% per population, a typical rate for desert plants that could indicate either low reproductive success or strong dormancy, e.g., Ref. [27]. Due to these natural challenges, we decided to subset our field data and include only plants that contributed seeds to the net house experiment. We believe that despite the reduction in sample size, the general trends observed in the field were similar, thereby allowing us to generate more coherent and reliable results that facilitate our understanding of the process between the two parts of the study. Another potential caveat may be that we only tested plasticity in response to water availability, while plants may also respond to various other habitat conditions. Undoubtedly, there is an infinite number of potential habitat conditions that could be examined. However, considering that arid conditions are primarily defined by limited precipitation, we deemed it logical to use irrigation as the primary factor affecting these plants in our experiments. 

In conclusion, plant species that are highly adapted to stressful conditions are typically expected to exhibit lower plasticity due to the association between local adaptation and phenotypic stability [6,7]. However, our findings challenge this notion by demonstrating that plants thriving in stressful yet highly variable conditions, such as the arid region, may employ phenotypic variability as a strategic adaptation to their habitat. These results contribute to the ongoing discussion on trait variability and its integral role in maintaining organismal success under stress [4,28]. Particularly, our findings suggest that the stress-induced variability hypothesis, e.g., Ref. [48], is a highly relevant framework. Furthermore, given the rapid changes in global conditions, understanding the potential responses of plants thriving in arid and warm conditions becomes highly pertinent [49]. These regions, often associated with marginal habitats, offer a valuable and convenient case study to gain deeper insights into the coping mechanisms of plants with respect to conditions that might become even more challenging in the future. 

## 4. Materials and Methods

### 4.1. Study Species

*A. hierochuntica* L., the True Rose of Jericho (Brassicaceae), is a winter desert annual that colonizes the driest zones across North Africa and Western Asia, mainly in Wadies and runnels [22,23]. The species varies in size, from 1 cm in height with as little as two flowers to 20 cm in height with a diameter of 50 cm and numerous flowers. *Anastatica hierochuntica* is a continuously flowering plant that produces flowers along its life cycle. The fruit of the species is a silique (pod) containing four seeds in two rows (upper and lower). The species is a lignified annual, with a unique hygrochastic mechanism in which only a few seeds are released in each rainfall event and the others are left as an aerial seed bank on the dead skeleton of the mother plant for several years [22,27]. In events of sufficient rainfall, the plant skeleton uncurls its branches to expose the fruits, allowing the raindrops to release some of the seeds. The seeds fall to the wet ground next to the mother plant, adhere to the moist soil surface, and germinate within six to ten hours [50]. Additionally, Friedman et al. [22] showed that there are differences in the force required for dispersal between the peripheral and central fruits of the plant. Because *A. hierochuntica* is a continuously flowering plant, this finding suggested that the mother plant invests differently in offspring at different developmental stages and ages. 

In addition to the highly adapted dispersal mechanism, the species is known to be tolerant to typical stress factors in arid environments, such as high soil salinity and radiation loads [26,51,52].

### 4.2. Population Survey in the Field

Three sites located in the Negev Desert, Israel, were selected for this study in winter 2019. The Negev Desert, an arid to hyperarid region, is characterized by mild temperatures and limited rainfall events in winter and hot temperatures and dry weather in summer. The sites created an aridity gradient that was negatively correlated with a soil salinity gradient. *A. hierochuntica* was the dominant species in all the sites (Figure 6).

In each of the sites, we chose a 50 m × 50 m plot. Within the plots, we selected fifteen individual plants that were haphazardly distributed throughout the plot. The exact geographical location of each individual was recorded with a submetric GPS receiver (STONEX S5 GNSS receiver; Lissone, Italy). The individuals were marked and used for trait measurements in the field and later for seed collection for the net house experiment. The number of plant samples was constrained by population sizes and a highly limited growth period due to an unexpected heat spell. This also constrained our ability to collect all the field measurements at all sites. Nevertheless, we collected all the relevant environmental data and seeds for the net house experiment. 

### 4.3. Environmental Data

In addition to the average site conditions, we evaluated the local habitat conditions near each individual plant by translating the individual’s topographical position to a value representing the relative local elevation in which the individual grew. Specifically, we created digital elevation models (DEMs) for each site by using a drone with an RGB camera (DJI Phantom 3 Advanced; Shenzhen, China) based on photos taken 30–50 m above ground. The transformation of captured images to topographic models was conducted with the mapping software Pix4DMapper Pro v.3.2.23; [53]. By laying the exact coordinates of each sampled plant on the DEM, we could assess its relative topographic height in a more precise way. The relative topographic position of the plant was used as a substitute for direct information on water availability to the individual plants. This was required because a direct measurement, which will correctly reflect the amount of water available to the plants in the long-term, was highly difficult to acquire due to the highly unpredictable nature of the duration and occurrence of rainfall events in the study region. Moreover, because of the nature of the arid sites, most of the rainfall creates runoff water that flows in small runnels within the sites and accumulates in lower areas. Therefore, we utilized the nature of runoff and water distribution in the sites and assumed that the lower a plant was growing, the more water was available to the individual plant at the local scale. 

In addition, we measured the soil salinity in the site by systematically sampling soil every 5 m within the plot. Each sample consisted of 500 g of soil collected at a depth of 20 cm. The soil salinity in each point was assessed by diluting 20 g of soil from each sample with 20 mL of double distilled water, mixing it in a shaker (TOS-4030FD, mrc; Holon, Israel) for 12 h, and measuring the electric conductivity of the mixed solute by using a conductivity meter (CON 510, Eutech Instruments, Singapore). The electric conductivity (EC) data were combined with the geographical data to create a continuous soil salinity map by using kriging [54]. 

### 4.4. Functional Traits

Due to the harsh conditions of the study sites and their relative remoteness, we limited our choice of functional traits to nondestructive traits that could be easily measured without the need for lab equipment or a fast delivery to laboratory conditions. The traits chosen were the plant height and the number of leaves, both indications of plant size, a functional trait that is strongly correlated with plant fecundity and growth rate [55]. Height was measured in the sites with a standard measuring tape from the base of the stem near the ground to the highest point of the shoot. The measures were taken on the same day at each site at the peak of the growth season. Leaf counting was performed on the site on the same day as the height measurements. 

In two of the sites that differed in their aridity level (Timna and Shacharut, Figure 6), we managed to additionally collect three young leaves from each sampled plant and kept them in a dark cool box until reaching the lab and measuring two additional functional traits. All the leaves were processed in the lab on the same day of collection. 

One of the leaves was randomly used to measure the specific leaf area (SLA), which is associated with the growth rate. Based on the Pérez-Harguindeguy et al. [55] protocol, the image of each leaf was captured under a binocular with a microscope camera (6EIII, DeltaPix, Smorum, Denmark). The images were analyzed with ImageJ [56] to generate the leaf surface area. After measuring the surface area, the leaves were dried in an oven at 60 °C for 72 h and weighed. 

The two other leaves were used to measure electrolyte leakage, a trait that is associated with plant tolerance to arid conditions, because higher leakage is related to higher membrane damage and lower cell integrity due to water limitations [35] and to salt stress [57]. Electrolyte leakage was measured by comparing the differences in leaf conductivity before and after the leaf cells were physically damaged. The estimation was based on measuring the electric conductivity (initial EC) of a fresh leaf sunk in double distilled water and shaken for 12 h. A second measurement (max EC) was taken after breaking the leaf cell membrane in an Autoclave (Tuttnauer 5075 HS, Breda, The Netherlands), forcing ions out of the tissue. The relative EC was then calculated as (Initial EC)/(Max EC) × 100. The larger the value of the relative EC, the more tolerant the leaf is to stress. 

### 4.5. Net House Experiment

The net house experiment was designed to test whether the trait variability detected in the field can be attributed to phenotypic plasticity. We therefore used seeds collected from all the individuals (mothers) measured in the field. To compensate for the potential differences in offspring investment along the plant life, we collected fruits from the center and periphery of each individual sampled in the field. Because of the indications for different maternal effects within the plant, we further separated seeds from the upper and lower rows within a fruit. In total, we aimed to collect eight seeds from each mother plant (2 fruit positions × 2 seed positions within a fruit × 2 seeds). Because of their sensitivity to humidity, the seeds were kept in separate Eppendorf tubes with silica gel in dark controlled conditions until sowing.

We sowed the seeds in January 2020 in 3 L pots filled with a mixture of 33% potting soil and 66% local Loess soil. The pots were irrigated twice a week for 20 min throughout the experiment, using either 2 L/H drips or 1 L/H drips (for the high and low irrigation treatments, respectively). Overall, the experiment contained 2 irrigation treatments × 3 sites × 15 mothers per site × 4 replicates per individual (fruit positions × 2 seed positions within fruit) = 360 pots. The pots were positioned in a full random design in a net house located in the Sede Boker Campus of Ben-Gurion University (30°51′8.27″ N, 34°47′0.24″ E), Israel, where conditions were similar to the ones in the field sites.

### 4.6. Functional Traits

The plant height, number of leaves, and SLA were measured in the net house experiment in the same fashion that they were measured in the field. In addition, due to the nature of the net house experiment, we could also measure the dry biomass of each plant at the end of the experiment and count the number of fruits produced by each plant as a measure of plant fitness.

### 4.7. Statistical Analysis

#### 4.7.1. Population Survey in the Field

To identify the scale of the environmental conditions affecting the trait values of *A. hierochuntica,* we conducted a set of generalized linear mixed models (GLMMs). In this set, the value of the measured functional trait was used as the response variable, and the environmental measurements were used as fixed effects. We used the individual plant’s local elevation and the soil salinity (EC) near its location as the local environmental conditions within site and the site’s aridity value (the de Matronne index) as the site conditions together with the field site identity as a random term grouping all individuals per population. All the data were scaled and centered before analyses to prevent convergence problems due to different scales of measurement units and to facilitate comparisons of the results across terms and models. When testing for the effects of the number of leaves, we added individual plant height as a covariate to account for potential allometric relations between height and the number of leaves in plants. We used a normal distribution with a log link function for all the traits except for the number of leaves, where we used a Poisson distribution with a log link. The models were fitted by using the “lme4” package, and all the model assumptions, including the multicollinearity between variables and residual distribution fit, were tested [58]. Pairwise-comparison plots were generated by using the “ggstatsplot” package in R software V. 4.2.3 [59,60].

To answer the second question, we calculated Euclidian distances among all the plants for all the measured plant traits as well as for the explanatory variables, which were the habitat conditions measured for each plant. The dissimilarity distances in the specific trait values between each pair of plants were then used in a set of general linear mixed models (GLMMs), where the dissimilarity distances were modeled as a function of the dissimilarity distances in the corresponding habitat conditions. Interactions between dissimilarity distances in site-level variables (the aridity index) and in variables of local environmental conditions within site (the soil EC and local elevation) were added to the models. Two nested random variables were added to the model to account for individual plant identity and the population from which the plants originated.

#### 4.7.2. Net House Experiment

We calculated the mean trait values per mother plant and irrigation treatment for each of the five traits that were measured in the net house. These mean trait values were then used as response variables in a set of GLMMs (normal distribution and identity link), using the same environmental variables as measured in the field. In these models, each of the measured functional traits was used as a response variable. The environmental variables, the irrigation treatment, and their interactions were used as explanatory fixed effects, and the mother-plant identity nested within sites of origin was used as the random effects.

The plasticity index per individual was calculated based on the creation of reaction norms by running a set of simple linear regressions for all offspring of each mother plant, with the treatment as an explanatory variable and the standardized trait values as the response variables. The slopes of the regressions extracted from these analyses were then used as the response variables in a set of GLMMs, with the same fixed and random variables as in the previous analyses.

## Figures and Tables

**Figure 1 plants-13-00256-f001:**
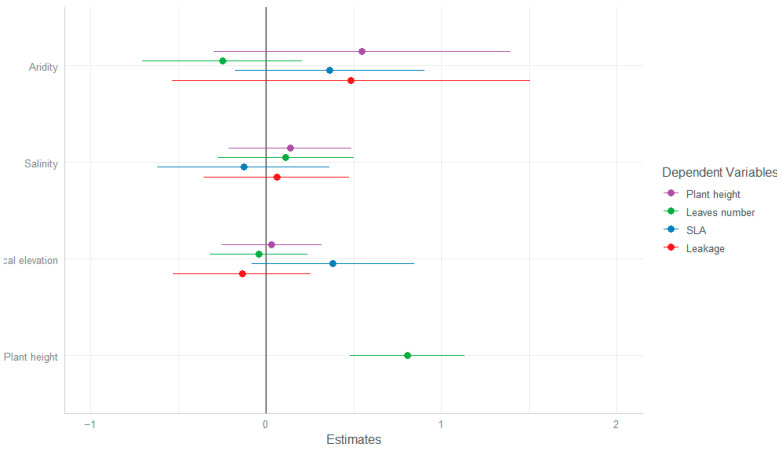
GLMM slope values (±SE) for the effects of between-site (aridity index) and within-site (salinity and local elevation) conditions on traits of *A. hierochuntica* individuals. Plant height was also added as a covariate in the analysis of number of leaves.

**Figure 2 plants-13-00256-f002:**
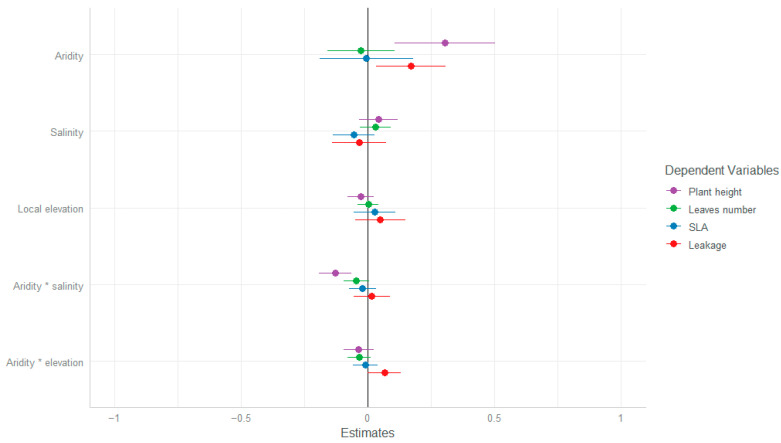
GLMM slope values (±SE) for the effects of dissimilarity in between-site (aridity index) and within-site (salinity, local elevation) habitat conditions on traits dissimilarity of *A. hierochuntica* individuals.

**Figure 3 plants-13-00256-f003:**
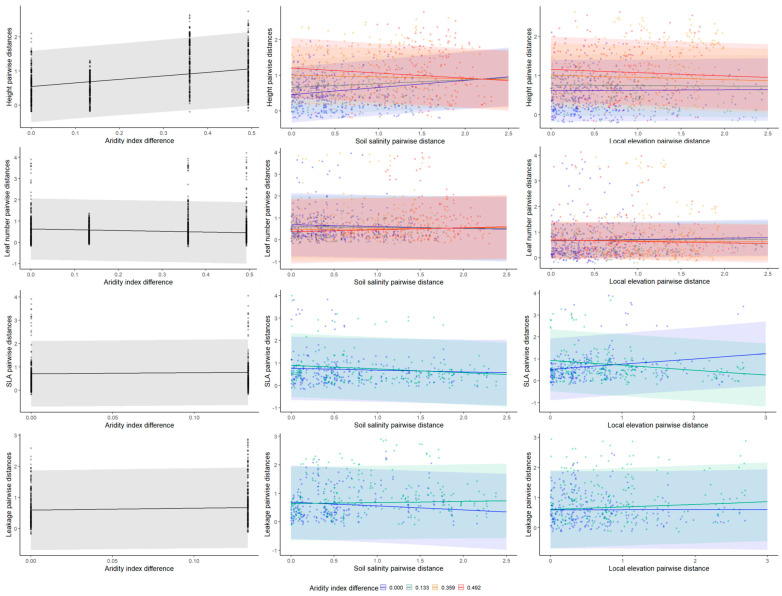
The effect of dissimilarity in between-site (aridity index) and within-site (salinity and local elevation) habitat conditions on trait dissimilarity of *A. hierochuntica* individuals. Note that the analysis is based on Euclidian distance (dissimilarity) values; thus, the larger the value, the less similar the two original measures that are being compared. Additionally, because aridity was measured at the population level, zero dissimilarity represents the within-population effect. Larger dissimilarity values represent the differences between the sites.

**Figure 4 plants-13-00256-f004:**
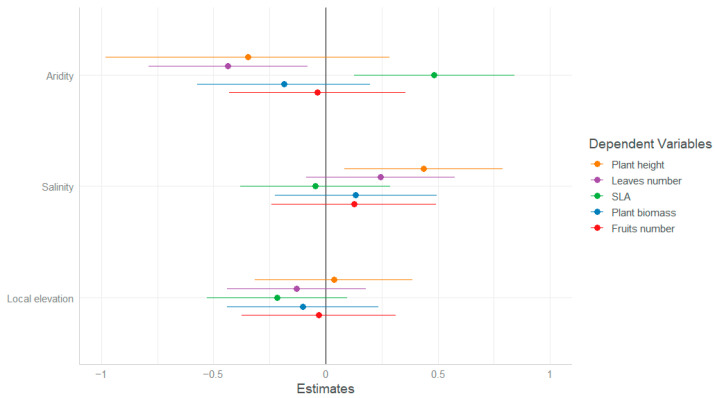
GLMM slope values (±SE) for the effects of between-site (aridity index) and within-site (salinity and local elevation) habitat conditions of the mother plants on plasticity indices based on different traits of *A. hierochuntica* individuals.

**Figure 5 plants-13-00256-f005:**
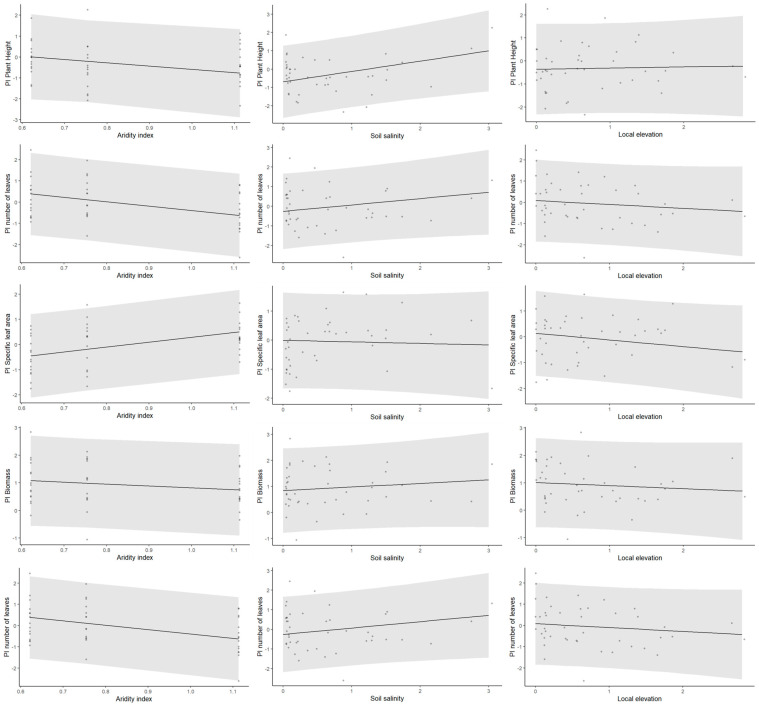
The effect of between-site (aridity index) and within-site (salinity and local elevation) habitat conditions on the change in plasticity index (PI, based on individual plant’s reaction norms) in *A. hierochuntica* individuals.

**Figure 6 plants-13-00256-f006:**
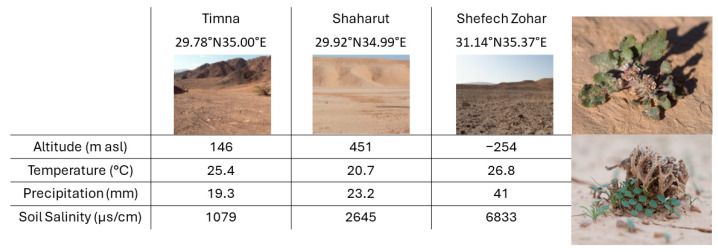
Study site information and photographs of the study plant species (adult in flowering and a dead plant with seedlings around it).

**Table 1 plants-13-00256-t001:** GLMM results (χ^2^ and df) for testing the potential effects of between-site (aridity index) and within-site (soil salinity and local elevation) conditions on trait values of *A. hierochuntica* individuals. *** represents a *p*-value smaller than 0.01.

		Height χ^2^(df)	Leaves Number χ^2^(df)	SLA χ^2^(df)	Electrolyte Leakage χ^2^(df)
** *Site scale* **	**Aridity**	1.7142(2)	1.2329(2)	1.8891(1)	1.0783(1)
** *Local scale* **	**Salinity**	0.6339(1)	0.3457(1)	0.2905(1)	0.0891(1)
	**Local elevation**	0.0515(1)	0.0835(1)	2.8729(1)	0.5262(1)
** *Covariates* **	**Height**		**24.7515 ***(1)**		
** *Conditional R^2^* **		0.662	0.456	0.118	0.505

**Table 2 plants-13-00256-t002:** GLMM results (χ^2^ with 1 df) for testing the potential effect of dissimilarity in between-site (aridity index) and within-site (soil salinity and local elevation) conditions on the dissimilarity in *A. hierochuntica* trait values (based on Euclidian distances). * represents 0.05 < *p*-value < 0.01, ** represents 0.01 < *p*-value < 0.001, and *** represents *p*-value < 0.001.

		*Plant Height*	*Leaves Number*	*SLA*	*Electrolyte Leakage*
** *Site scale* **	**Aridity**	**9.0136 ****	2.6397	0.0038	**5.6025 ***
** *Local scale* **	**Salinity**	0.0490	0.2562	1.7602	0.4499
	**Local elevation**	1.0811	0.0126	0.3554	1.6168
	**Aridity × salinity**	**15.0710 *****	3.3534	0.5997	0.1624
	**Aridity × local elevation**	1.4808	2.3731	0.1769	3.7699
** *Conditional R* ** ** ^2^ **		0.611	0.793	0.853	0.6730

**Table 3 plants-13-00256-t003:** GLMM results (χ^2^ with 1 df) for reaction norm of plants relative to the between-site (aridity index) and within-site (salinity and local elevation) habitat conditions of their mother plants. * represents 0.05 < *p*-value < 0.01, ** represents 0.01 < *p*-value < 0.001.

		*Height*	*Leaves Number*	*SLA*	*Biomass*	*Fruits Number*
** *Site scale* **	**Aridity**	1.2366	**6.1838 ****	**7.5182 ****	0.9555	0.0389
** *Local scale* **	**Salinity**	**6.3152 ***	2.2407	0.0803	0.5735	0.4884
	**Local elevation**	0.0430	0.7227	1.9718	0.3787	0.0336
** *Conditional R* ** ** ^2^ **		0.280	0.187	0.175	0.047	0.013

## Data Availability

The data presented in this study are available upon request from the corresponding author. The data are not publicly available due to privacy.

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
