# Peer review of "Stress Induces Trait Variability across Multiple Spatial Scales in the Arid Annual Plant Anastatica hierochuntica"

_plants, 2024, doi:10.3390/plants13020256_

Round 1

Reviewer 1 Report

Comments and Suggestions for Authors

I consider research to be very important, also in view of the changing conditions affecting the whole world. The results may help in the future in adapting to climate change, which we are also experiencing in areas of Central Europe. It is clear that results of this type can help in efforts to green the desert. There are, of course, many questions that remain to be answered, so I hope that the authors will continue to pursue this topic. 

The comments I have on the submitted paper are stylistic only and in no way detract from the Results or Discussion. I appreciate the appropriate use of the GLMM, including such detail as the addition of plant heights as covariates.

The Introduction is clear and concise, and if the conditions of the journal permit, I would move the section on Anastatica hierochuntica, now in the Materials and Methods chapter, to the Introduction. I think the text will then be smoother, also considering that the M&M chapter is after the Discussion.

Note that Latin names should be italicized, as is standard.

The Results and Discussion are written very well and concisely.

The M&M chapter is not described completely well. It took me a while to understand which variables were actually used. I think it is unnecessary to justify why some variables were not used and thus "loosen" the story of the experiment. However, if you used some sort of model to interpolate the humidity data based on topographic location, it would be useful to describe that. 

A reminder for further research: it would be interesting to include seed production or 1000 seed weight calculated from harvest in the model.

I would recommend omitting the parts about what you didn't do but wanted to do (it can't be changed or fixed and is therefore pointless to talk about). See Line 401 onwards. If electrolytes were not measured, it is useless to talk about it. Although of course I understand the reasons you mention it. Measuring electrolytes helped you to determine the habitat gradient - you are confusing seeds from different conditions. 

The statistical methods chapter is described clearly, concisely and understandably.

Overall, I found the article very successful and useful. After editing, I recommend it for publication in Plants.

Author Response

Detailed response can be found in the attached file

Reviewer 2 Report

Comments and Suggestions for Authors

The manuscript investigates traits of Anastatica hierochuntica between and within different arid sites in Negev Desert. They found that plant species that are highly adapted to stressful conditions are in stressful yet highly variable conditions, plants employ phenotypic variability as a strategic adaptation to their habitat.

The manuscript is interesting, well written and scientific sound. I have some minor concerns:

TITLE: I suggest to report the species name, i.e. : Stress induces trait variability across multiple spatial scales in the  arid annual plant Anastatica hierochuntica L.

L 79: here and in the entire text, please pay attention on the use of italics when reporting scientific latin names

L 291: authorship is not in italics, in any case the authorship should be reported only when the taxon is first mentioned

L 310 I suggest inserting a figure here on the sites, e.g. three pictures, one for each site and a small map to locate them geographically

L 335: please explain why a DEM was created from drone's camera to extrapolate the elevation, while it was sufficient to record it directly by the GPS

L 338 please check the comma before the citation 31

L 406 I suggest testing and then reporting (expecially if r >0.70) collinearity among explanatory variables used in each model

L 522 check taxa names, also in references, e.g. Ref. 24-25, names in italics and species with lower case

Author Response

(The authors gave the same response as above.)

Reviewer 3 Report

Comments and Suggestions for Authors

Dear authors,

You are presenting an interesting paper based on trait variability across multiple spatial scales in an arid annual plant induced by stress. I like the overall idea and the methodology of your paper that gives interesting results. The manuscript is well written and structured. Discussion could be enriched with more comparisons of your results with those of other studies investigating stress induced trait variability in plant populations. This version of the manuscript could be much improved.  I believe that a revision of the MS is needed so that the science presented, which is interesting, can be communicated more effectively.

Please find below my suggestions for the manuscript.

All over the MS: Anastatica hierochuntica and A. hierochuntica, please write it in italics.

Page 3-4, Results, 2.1. Population survey in the field: Text is missing before Table 1, needed also to refer to Table 1 and to Figure 1. Additionally, Table 2 and Figure 2 should be mentioned in the text before their presence.

Page 6, line 165: I think Table 3 is not mentioned in the text. Please add it in text before the Table.

Page 10, line 289, M & Ms: The text could be shortened enough and move more detailed information to a Supplementary file. Please also check numbering of references (for example, I think reference 27 is presented for the first time here).

I hope that my comments will be helpful.

Author Response

(The authors gave the same response as above.)

Round 2

Reviewer 3 Report

Comments and Suggestions for Authors

Dear authors,

The revised version of your manuscript is much improved.